# Prevalence, patterns, and determinants of multimorbidity in South Africa: Insights from a nationally representative survey

Matthew Hazell[1]*, Andre Pascal Kengne[2,3], Paramjit Gill[1], Dylan Taylor[1], Olalekan Uthman[1]

1 Warwick Applied Health, Warwick Medical School, University of Warwick, Coventry, United Kingdom, 2 Department of Medicine, University of Cape Town, Cape Town, South Africa, 3 Non-Communicable Diseases Research Unit, South African Medical Research Council, Cape Town, South Africa

* Matthew.Hazell@warwick.ac.uk

## Abstract

### Background

Multimorbidity in Sub-Saharan Africa is under researched and includes distinct disease combinations to those seen in high income countries. The aim of this study was to determine the prevalence and distribution of multimorbidity in South Africa, as well as the associated individual, area-level and contextual factors.

### Methodology

Multilevel logistic regression analyses were conducted on nationally representative 2016 South Africa Demographic Health Survey Data. The sample included 5,592 individuals (level 1) living in 691 neighbourhoods (level 2).

### Principal findings

Multimorbidity was present in 45.3% of the study population, ranging from 35.6% in Limpopo to 52.1% in Eastern Cape. Hypertension was the most prevalent condition (46.4%) followed by diabetes (22.6%). Individuals aged 65–95 had 11.57 times higher odds (95% CI 8.50-15.74) of multimorbidity compared to those aged 15–24. Women had nearly twice the odds of men (OR 1.95, 95% CI 1.68-1.27). Formerly married individuals had 1.63 times higher odds (95% CI 1.32-2.02) than never married. Compared to Black Africans, White individuals had 44% lower odds (OR 0.56, 95% CI 0.39-0.82) and those of mixed ethnicity had 31% lower odds (OR 0.69, 95% CI 0.51-0.92). Obesity increased the odds by 38% (OR 1.38, 95% CI 1.17-1.64) and occupational smoke exposure by 26% (OR 1.26, 95% CI 1.07-1.49). There was variation in multimorbidity at the neighbourhood level, with 2.9% of the variation attributable to contextual factors in the empty model. The median odds ratio was 1.35, indicating substantially higher odds of multimorbidity if an individual moved to a higher risk neighbourhood.

**Data availability statement:** SADHS data are available to download from: https://preview.dhsprogram.com/data/available-datasets.cfm.

**Funding:** This work was supported by the National Institute for Health and Care Research (NIHR302654 to MH). The funders had no role in study design, data collection and analysis, decision to publish, or preparation of the manuscript. The views expressed in this publication are those of the author(s) and not necessarily those of the NIHR, NHS or the UK Department of Health and Social Care.

**Competing interests:** The authors have declared that no competing interests exist.

## Conclusions

This study found a high burden of multimorbidity in South Africa patterned by demographic, socioeconomic, lifestyle and contextual factors. The results highlight the need for multilevel strategies to reduce multimorbidity and its inequities by addressing individual risk factors as well as neighbourhood-level determinants of health.

## Background

Individuals living with multimorbidity, defined as the co-existence of two or more chronic diseases, have a reduced quality of life, increased need to utilise healthcare, and greater mortality [1–5]. Approximately 37% of the global population is believed to be multimorbid, with the burden primarily on the elderly [6]. Nevertheless, multimorbidity remains under researched compared to single conditions – this is particularly notable in low- and middle-income countries (LMICs) where only 5% of multimorbidity research originates [7].

Multimorbidity is increasing in LMICs, particularly in Sub-Saharan Africa (SSA), due to population ageing, lifestyle changes, and a changing climate and environment [6]. Additionally, multimorbidity in SSA is distinct from that of high-income countries due to the high prevalence of persistent infectious diseases (including human immunodeficiency virus (HIV) and tuberculosis), maternal and neonatal diseases, and injury related diseases co-occurring with non-communicable diseases (NCDs) [8]. Concerningly, reports have also suggested that multimorbidity is appearing at younger ages in SSA [8]. Therefore, research within this context is required to inform the specific healthcare management of multimorbidity in SSA.

South Africa is an upper-middle-income country within SSA, and one of its most urbanised and developed nations. Many health inequalities exist in South Africa across socioeconomic, ethnic, and geographical divisions [9,10]. These inequalities are partly a legacy of the Apartheid era, which divided the population by race and deprived most South Africans of basic human rights, and the high levels of unemployment and urbanisation seen since the advent of democracy in 1994 [11]. Indeed, poorer groups are reported to experience a higher burden of NCDs, and a higher prevalence of NCD risk factors including binge drinking and obesity, despite NCDs being viewed as diseases of affluence [12].

There is considerable heterogeneity in prevalence estimates of multimorbidity in South Africa. A recent systematic review found estimates ranging from 3-23% in studies with younger people, and 30–88% in older adults [13]. This is due to contrasting study designs and differing definitions of multimorbidity: definitions that capture more chronic conditions will result in a higher multimorbidity prevalence. Of the single conditions that make up multimorbidity, hypertension, anaemia, and HIV are believed to be the most prevalent [14].

Identified individual-level risk factors for multimorbidity in South Africa include sex, age, area of residence, occupation, education, income, marital status and body mass index (BMI) [13,15,16]. However, the literature is sparse and contradictory.

The role of an individual's area of residence on multimorbidity, which has been demonstrated in Ghana, is likely also important in South Africa and intra-context correlation needs to be considered [17]. Higher burdens of multimorbidity have been reported in parts of KwaZulu-Natal and Eastern Cape and lower burdens in the provinces of Limpopo and Mpumalanga [10]. However, there has been no research into the extent of variation of multimorbidity by area in South Africa which is important to understand the influence of specific contexts on multimorbidity.

This study aims to determine the prevalence and distribution of multimorbidity in South Africa. This research will inform policy makers to allocate resources appropriately, to develop prevention programmes and manage multimorbidity.

## Materials and methods

### Ethics statement

The anonymised datasets with necessary approvals were obtained from the Demographic and Health Survey (DHS) programme (https://www.dhsprogram.com/data/) for this secondary analysis, and no further ethical clearance was required. Data were accessed on 24/10/2023 and authors had no access to information that could identify individual participants during or after data collection. All participants of the South Africa Demographic and Health Survey (SADHS) 2016 completed consent forms.

### Study design and data sources

SADHS 2016 is a nationally representative cross-sectional household survey that provides information on demographic and health indicators. The DHS programme has assisted with over 350 nationally representative household surveys across 90 countries since 1984 [18]. Surveys are usually conducted every five years per country, with topics tailored to relevant population health issues. The DHS is an important source of data for policy making, monitoring and evaluation in many LMICs. The SADHS 2016 followed a stratified two-stage sample design. Seven hundred and fifty primary sampling units (PSUs) were selected from the 26 sampling strata, based on urban, traditional and rural areas for each of the nine provinces in South Africa (there was no strata for traditional areas in Western Cape). A fixed number of 20 dwelling units (DUs) were randomly selected from each of the PSUs. This design permits estimates of key variables for the country, for each of the nine provinces, and of urban, rural and traditional areas. Data collection took place over six months, from 27 June 2016 to 4 November 2016.

### Study population

Men and women aged 18 and over were eligible for inclusion into this study if they contributed data to SADHS 2016 adult health modules. Individuals under 18 were excluded to ensure comparability to the literature. Individuals were also excluded from the study if they were missing information on the multimorbidity outcome.

### Variables

**Outcome variable.** Multimorbidity is measured by counting the number of co-existing chronic conditions, with a cut-off of at two or more conditions [19]. Twelve current chronic diseases contributed to the binary multimorbidity outcome (multimorbidity or no multimorbidity): tuberculosis, hypertension, stroke, high blood cholesterol, anaemia, chronic bronchitis, diabetes, asthma, cancer, heart disease, HIV, and chronic pain. Further information on how these variables were derived and coded can be found in S1 Table in the Supplementary file.

**Explanatory variables.** Included explanatory variables were informed by literature review and a-priori reasoning, all were self-reported or derived from self-reported variables. The socioeconomic variables included were household wealth index, education level, occupational status, health insurance and marital status. Individual level health variables included BMI (missing for women who were pregnant or had been in the last two months), dietary health, sugary drink intake,

smoking status, alcohol drinking and exposure to smoke at work. Two variables representing access to old and new media, respectively, were included as a proxy for access to health information. Finally, variables for age, sex and ethnicity were also included. Those who answered 'don't know' were recorded as missing information on that variable.

Neighbourhoods were defined as respondents from clusters of households which serve as the PSU within the DHS. Poverty, rurality, and unemployment level were chosen as the neighbourhood and province level explanatory variables, with illiteracy only investigated at the neighbourhood level. These were defined as the proportion of individuals living in the most deprived group for each neighbourhood and province (proportion in the lowest wealth category, living rural or traditional, unemployed and illiterate). This was split into three categories (low, medium and high), with neighbourhood rurality as a binary variable, and calculated with the larger adult-health sample (N = 9,512) to include more contextual information. Further information on how these variables were derived and coded can be found in S2 Table in the Supplementary file.

## Statistical methods

**Descriptive statistics.** To describe and compare the characteristics of individuals in the study population with and without multimorbidity, descriptive statistics were produced which summarised the included covariates in those with and without multimorbidity. A description of the data by province was also produced. To identify the conditions contributing most to the burden of multimorbidity the prevalence of each chronic condition in the study population was recorded.

**Model development and variable selection:** To reduce the dimensionality of the model and avoid overfitting, we employed LASSO (Least Absolute Shrinkage and Selection Operator) regression for variable selection [20]. LASSO is a penalized regression technique that shrinks the coefficients of less important variables to zero, thereby selecting only the most relevant predictors. This approach is particularly useful when dealing with a large number of covariates, as it minimizes multicollinearity and improves model interpretability. The LASSO model was fitted using the EBIC (Extended Bayesian Information Criterion) to select the optimal lambda value, which balances model fit and complexity. The final model included the following variables, selected based on their contribution to explaining the variance in multimorbidity: Age category, sex, wealth index, education attainment, marital status, ethnicity, body mass index, workplace smoke exposure, new media access and region. The LASSO regression results indicated that these variables were the most significant predictors of multimorbidity, with minimal loss of predictive accuracy

**Multilevel approach.** To analyse the individual/ household (level 1) and neighbourhood (level 2) level factors associated with multimorbidity, multivariable multilevel logistic regression models were produced. Multilevel analysis has been utilised in much prior social epidemiology research when investigating hierarchical data and considers the individual probability of the outcome to be statistically dependent on area of residence [21–30]. Two models were fitted. Model 1 contained no covariates to demonstrate the variance in the outcome variable attributed to clustering at the neighbourhood level. After variable selection using LASSO, the model 2 was simplified to include only the most relevant predictors, reducing the risk of overfitting and multicollinearity.

Adjusted odds ratios with 95% confidence intervals were reported for fixed effects associations between individual, neighbourhood and province level variables and the outcome. Measures of area-level variance from random effects, were the median odds ratio (MOR) and variance partition coefficient (VPC). The MOR estimates the variance in multimorbidity expressed as an odds ratio attributed to neighbourhood and province contexts and represents the extent to which the individual probability of multimorbidity is determined by residing in a neighbourhood and province. The MOR can be conceptualised as the median increased risk of multimorbidity from moving to an area with a higher risk [27,31]. The VPC represents the proportion of response variance at the neighbourhood and province levels of the model. A Chi-squared test was used to determine the strength of evidence for variation at the neighbourhood and province level.

All analyses were conducted using STATA 18 and descriptive analyses were adjusted for sample weight, stratification and clustering as per DHS recommendations.

## Results

### Characteristics of the study population

The study sample comprised 5,592 participants, with 2,533 (45.3%) individuals identified as having multimorbidity, defined as the co-occurrence of two or more chronic conditions. The prevalence of individual chronic conditions varied significantly (Fig 1), with hypertension being the most common (46.4%), followed by diabetes (22.6%), anaemia (22.0%), chronic pain (21.2%), and HIV (20.9%). Less prevalent conditions included high blood cholesterol (4.2%), heart attack (4.1%), asthma (4.0%), chronic bronchitis (1.6%), tuberculosis (1.5%), stroke (1.4%), and cancer (1.2%).

The distribution of multimorbidity varied across demographic and socioeconomic characteristics (Table 1). Age was a strong predictor of multimorbidity, with prevalence increasing sharply with advancing age. Only 15.5% of individuals aged 15–24 years had multimorbidity, compared to 71.8% of those aged 65–95 years (p<0.001). Sex differences were also notable, with women more likely to have multimorbidity than men (52.1% vs. 33.7%, p<0.001).

Socioeconomic factors, such as educational attainment and wealth index, were also associated with multimorbidity. Individuals with no formal education had the highest prevalence of multimorbidity (64.7%), while those with higher education had the lowest (33.8%, p<0.001). Access to new mass media, a proxy for health information and awareness, was significantly associated with lower multimorbidity prevalence. Individuals with high media access had a 27.5% prevalence of multimorbidity, compared to 52.4% among those with low access (p<0.001).

Body mass index (BMI) was strongly associated with multimorbidity, with obese individuals having the highest prevalence (59.4%, p<0.001). Exposure to workplace smoke, dust, or fumes was weakly associated with multimorbidity (p=0.073).

Geographic disparities were also evident, with significant regional variations in multimorbidity prevalence (p<0.001). The Eastern Cape had the highest prevalence (52.1%), while Limpopo had the lowest (35.6%).

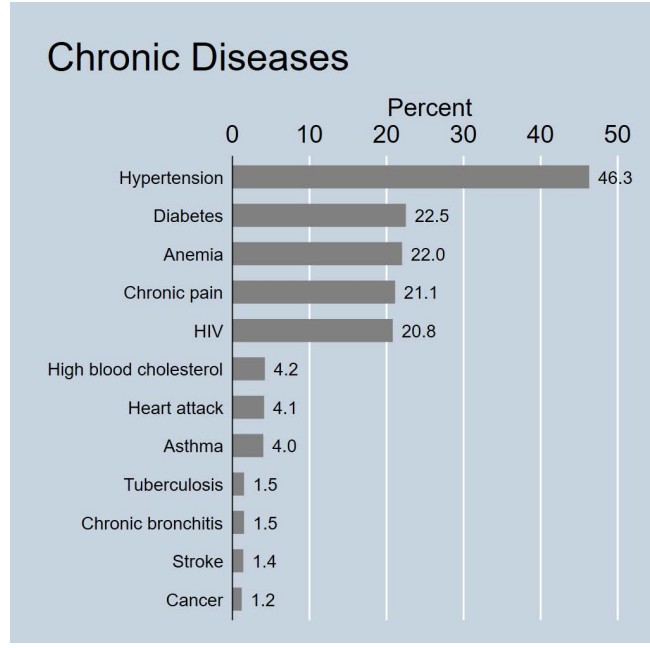

**Fig 1 Prevalence of individual chronic diseases in the study population.** Adjusted for sample weight, stratification and clustering.

**Table 1. Characteristics of the study population by multimorbidity status.**

| | Total | Multimorbidity | | p-value |
|---|---|---|---|---|
| | | No | Yes | |
| | N = 5,592 | N = 3,059 | N = 2,533 | |
| Current age | | | | <0.001 |
| 15–24 | 1,121 (20.0) | 947 (84.5) | 174 (15.5) | |
| 25–34 | 1,274 (22.8) | 868 (68.1) | 406 (31.9) | |
| 35–44 | 924 (16.5) | 484 (52.4) | 440 (47.6) | |
| 45–54 | 809 (14.5) | 329 (40.7) | 480 (59.3) | |
| 55–64 | 702 (12.6) | 216 (30.8) | 486 (69.2) | |
| 65–95 | 762 (13.6) | 215 (28.2) | 547 (71.8) | |
| Sex | | | | <0.001 |
| Woman | 3,526 (63.0) | 1,690 (47.9) | 1,836 (52.1) | |
| Man | 2,066 (37.0) | 1,369 (66.3) | 697 (33.7) | |
| Wealth index | | | | 0.92 |
| Poorest | 1,229 (22.0) | 678 (55.2) | 551 (44.8) | |
| Poorer | 1,217 (21.8) | 656 (53.9) | 561 (46.1) | |
| Middle | 1,326 (23.7) | 730 (55.1) | 596 (44.9) | |
| Richer | 1,097 (19.6) | 593 (54.1) | 504 (45.9) | |
| Richest | 723 (12.9) | 402 (55.6) | 321 (44.4) | |
| Educational level | | | | <0.001 |
| No education | 535 (9.6) | 189 (35.3) | 346 (64.7) | |
| Primary | 1,122 (20.1) | 464 (41.4) | 658 (58.6) | |
| Secondary | 3,461 (61.9) | 2,092 (60.4) | 1,369 (39.6) | |
| Higher | 474 (8.5) | 314 (66.2) | 160 (33.8) | |
| Marital Status | | | | <0.001 |
| Never married | 2,667 (47.7) | 1,768 (66.3) | 899 (33.7) | |
| Formerly in union | 810 (14.5) | 236 (29.1) | 574 (70.9) | |
| Currently in union | 2,115 (37.8) | 1,055 (49.9) | 1,060 (50.1) | |
| Ethnicity | | | | 0.69 |
| Black/African | 4,900 (87.6) | 2,678 (54.7) | 2,222 (45.3) | |
| White | 213 (3.8) | 123 (57.7) | 90 (42.3) | |
| Mixed | 430 (7.7) | 234 (54.4) | 196 (45.6) | |
| Asian/other | 49 (0.9) | 24 (49.0) | 25 (51.0) | |
| Body mass index | | | | <0.001 |
| Underweight | 284 (5.1) | 185 (65.1) | 99 (34.9) | |
| Normal weight | 2,106 (37.7) | 1,368 (65.0) | 738 (35.0) | |
| Overweight | 1,381 (24.7) | 735 (53.2) | 646 (46.8) | |
| Obese | 1,667 (29.8) | 676 (40.6) | 991 (59.4) | |
| Missing | 154 (2.8) | 95 (61.7) | 59 (38.3) | |
| Exposure to workplace smoke, dust, fumes | | | | 0.073 |
| Not exposed | 4,392 (78.5) | 2,430 (55.3) | 1,962 (44.7) | |
| Exposed | 1,200 (21.5) | 629 (52.4) | 571 (47.6) | |
| New mass media access | | | | <0.001 |
| Low | 651 (11.6) | 310 (47.6) | 341 (52.4) | |
| Medium | 3,237 (57.9) | 1,514 (46.8) | 1,723 (53.2) | |
| High | 1,704 (30.5) | 1,235 (72.5) | 469 (27.5) | |

*(Continued)*

**Table 1.** (Continued)

| | Total | Multimorbidity | | p-value |
|---|---|---|---|---|
| | | No | Yes | |
| | N = 5,592 | N = 3,059 | N = 2,533 | |
| Region | | | | <0.001 |
| Western cape | 343 (6.1) | 184 (53.6) | 159 (46.4) | |
| Eastern cape | 879 (15.7) | 421 (47.9) | 458 (52.1) | |
| Northern cape | 375 (6.7) | 193 (51.5) | 182 (48.5) | |
| Free state | 687 (12.3) | 369 (53.7) | 318 (46.3) | |
| KwaZulu-Natal | 787 (14.1) | 415 (52.7) | 372 (47.3) | |
| Northwest | 798 (14.3) | 469 (58.8) | 329 (41.2) | |
| Gauteng | 465 (8.3) | 288 (61.9) | 177 (38.1) | |
| Mpumalanga | 576 (10.3) | 281 (48.8) | 295 (51.2) | |
| Limpopo | 682 (12.2) | 439 (64.4) | 243 (35.6) | |

## Measures of association - Fixed effects

The multilevel logistic regression analysis revealed several significant factors associated with multimorbidity in South Africa (Table 2). Age was a strong predictor of multimorbidity, with the odds increasing progressively with older age groups. Compared to individuals aged 15–24 years (reference group), those aged 25–34 years had 2.20 times higher odds of multimorbidity (OR = 2.20, 95% CI: 1.77–2.75, p < 0.001). This trend continued, with odds ratios rising to 4.14 (95% CI: 3.25–5.28, p < 0.001) for ages 35–44, 6.20 (95% CI: 4.77–8.07, p < 0.001) for ages 45–54, 10.31 (95% CI: 7.70–13.81, p < 0.001) for ages 55–64, and 11.57 (95% CI: 8.50–15.74, p < 0.001) for ages 65–95. Women had significantly higher odds of multimorbidity compared to men (OR = 1.95, 95% CI: 1.68-2.27, p < 0.001). The wealth index did not show significant associations with multimorbidity across its categories (all p-values > 0.05). Similarly, educational attainment was not significantly associated with multimorbidity after adjusting for other variables. For example, individuals with higher education had odds of multimorbidity similar to those with no education (OR = 0.89, 95% CI: 0.63–1.26, p = 0.505). Marital status was associated with multimorbidity, particularly among individuals formerly in a union (e.g., divorced, widowed, or separated). Compared to those never married, individuals formerly in a union had 1.63 times higher odds of multimorbidity (OR = 1.63, 95% CI: 1.32–2.02, p < 0.001). In contrast, those currently in a union did not show significantly different odds compared to the never-married group (OR = 1.08, 95% CI: 0.93–1.26, p = 0.302).

Ethnicity was significantly associated with multimorbidity. White individuals had 44% lower odds of multimorbidity compared to Black/African individuals (OR = 0.56, 95% CI: 0.39–0.82, p = 0.002). Similarly, individuals of mixed ethnicity had 31% lower odds (OR = 0.69, 95% CI: 0.51–0.92, p = 0.013). No significant differences were observed for Asian/Other ethnic groups (OR = 1.03, 95% CI: 0.52–2.06, p = 0.926). Obesity was strongly associated with multimorbidity. Obese individuals had 1.38 times higher odds of multimorbidity compared to those with normal weight (OR = 1.38, 95% CI: 1.17–1.64, p < 0.001). Overweight individuals did not show significantly different odds (OR = 1.09, 95% CI: 0.92–1.28, p = 0.327), nor did underweight individuals (OR = 1.08, 95% CI: 0.80–1.46, p = 0.628). Exposure to workplace smoke, dust, or fumes was associated with a 26% increase in the odds of multimorbidity (OR = 1.26, 95% CI: 1.07–1.49, p = 0.005). Access to new mass media was weakly associated with multimorbidity. Individuals with medium access had 21% higher odds of multimorbidity compared to those with low access (OR = 1.21, 95% CI: 0.99–1.49, p = 0.066), though this did not reach statistical significance. High media access was not significantly associated with multimorbidity (OR = 0.95, 95% CI: 0.73–1.22, p = 0.673). Significant regional variations in multimorbidity were observed. Compared to the Western Cape (reference group), individuals in Limpopo had 54% lower odds of multimorbidity (OR = 0.46, 95% CI: 0.32–0.67, p < 0.001), while those in the North West had 33% lower odds (OR = 0.67, 95% CI: 0.47–0.95, p = 0.025). No other regions showed significant differences.

**Table 2. Individual, neighbourhood and province-level factors associated with multimorbidity identified by multilevel logistic regression models.**

| Variable | Empty model (Model 1) | | Fully adjusted model (Model 2) | |
|---|---|---|---|---|
| | OR (95% CI) | p-value | OR (95% CI) | p-value |
| Fixed effect model | | | | |
| Current age | | | | |
| 15–24 | | | 1 (reference) | |
| 25–34 | | | **2.20 (1.77 to 2.75)** | 0.000 |
| 35–44 | | | **4.14 (3.25 to 5.28)** | 0.000 |
| 45–54 | | | **6.20 (4.77 to 8.07)** | 0.000 |
| 55–64 | | | **10.31 (7.70 to 13.81)** | 0.000 |
| 65–95 | | | **11.57 (8.50 to 15.74)** | 0.000 |
| Sex | | | | |
| Man | | | 1 (reference) | |
| Woman | | | **1.95 (1.68 to 2.27)** | 0.000 |
| Wealth index | | | | |
| Poorest | | | 1 (reference) | |
| Poorer | | | 1.21 (0.99 to 1.47) | 0.058 |
| Middle | | | 1.10 (0.90 to 1.34) | 0.366 |
| Richer | | | 1.05 (0.84 to 1.31) | 0.670 |
| Richest | | | 0.95 (0.72 to 1.27) | 0.747 |
| Educational level | | | | |
| No education | | | 1 (reference) | |
| Primary | | | 1.18 (0.92 to 1.51) | 0.195 |
| Secondary | | | 1.20 (0.94 to 1.55) | 0.145 |
| Higher | | | 0.89 (0.63 to 1.26) | 0.505 |
| Marital status | | | | |
| Never married | | | 1 (reference) | |
| Formerly in union | | | **1.63 (1.32 to 2.02)** | 0.000 |
| Currently in union | | | 1.08 (0.93 to 1.26) | 0.302 |
| Ethnicity | | | | |
| Black/African | | | 1 (reference) | |
| White | | | **0.56 (0.39 to 0.82)** | 0.002 |
| Mixed | | | **0.69 (0.51 to 0.92)** | 0.013 |
| Asian/Other | | | 1.03 (0.52 to 2.06) | 0.926 |
| Body mass index | | | | |
| Underweight | | | 1.08 (0.80 to 1.46) | 0.628 |
| Normal weight | | | 1 (reference) | |
| Overweight | | | 1.09 (0.92 to 1.28) | 0.327 |
| Obese | | | **1.38 (1.17 to 1.64)** | 0.000 |
| Exposure to workplace smoke, dust, fumes | | | | |
| Not exposed | | | 1 (reference) | |
| Exposed | | | **1.26 (1.07 to 1.49)** | 0.005 |
| New mass media access | | | | |
| Low | | | 1 (reference) | |
| Medium | | | 1.21 (0.99 to 1.49) | 0.066 |
| High | | | 0.95 (0.73 to 1.22) | 0.673 |

*(Continued)*

**Table 2.** (Continued)

| Variable | Empty model (Model 1) | | Fully adjusted model (Model 2) | |
|---|---|---|---|---|
| | OR (95% CI) | p-value | OR (95% CI) | p-value |
| Region | | | | |
| Eastern Cape | | | 1.14 (0.80 to 1.61) | 0.468 |
| Northern Cape | | | 1.05 (0.73 to 1.51) | 0.781 |
| Free State | | | 0.81 (0.57 to 1.16) | 0.252 |
| Kwazulu-Natal | | | 1.02 (0.72 to 1.46) | 0.894 |
| North West | | | **0.67 (0.47 to 0.95)** | 0.025 |
| Gauteng | | | 0.71 (0.49 to 1.03) | 0.075 |
| Mpumalanga | | | 1.25 (0.86 to 1.82) | 0.236 |
| Limpopo | | | **0.46 (0.32 to 0.67)** | 0.000 |
| | | | | |
| Random effects model | | | | |
| Neighbourhood level | | | | |
| Variance (95% CI) | 0.10 (0.05 to 0.20) | | 0.08 (0.03 to 0.22) | |
| VPC (95% CI) | 2.9 (1.5 to 5.7) | | 2.4 (0.9 to 6.3) | |
| Explained variation | (reference) | | 16.4 (-11.1 to 37.0) | |
| MOR (95% CI) | 1.35 (1.23 to 1.53) | | 1.32 (1.18 to 1.57) | |
| | | | | |
| | | | | |

OR- odds ratio. CI-confidence Interval. N-number. Ref- reference group. VPC- variance partition coefficient. MOR- median odds ratio.

### Measures of variations - Random effects

As shown in Table 2, the multilevel analysis revealed significant variation in the odds of multimorbidity at the neighbourhood level, highlighting the influence of contextual factors on chronic disease burden. In the empty model (Model 1), the variance in multimorbidity odds across neighbourhoods was 0.10 (95% CI: 0.05 to 0.20), indicating substantial heterogeneity between neighbourhoods. The variance partition coefficient (VPC) showed that 2.9% (95% CI: 1.5 to 5.7) of the total variation in multi-morbidity could be attributed to neighbourhood-level factors. This suggests that contextual characteristics, such as healthcare access, environmental conditions, or social determinants of health, play a role in shaping the distribution of multimorbidity.

The median odds ratio (MOR) for the empty model was 1.35 (95% CI: 1.23 to 1.53), indicating that if an individual moved to a neighbourhood with a higher risk of multimorbidity, their median odds of being multimorbid would increase by 1.35-fold. This further underscores the importance of neighbourhood-level factors in influencing chronic disease outcomes.

In the fully adjusted model (Model 2), which included individual- and neighbourhood-level covariates, the neighbourhood-level variance decreased slightly to 0.08 (95% CI: 0.03 to 0.22), with a corresponding VPC of 2.4% (95% CI: 0.9 to 6.3). The explained variation at the neighbourhood level was 16.4% (95% CI: -11.1 to 37.0), indicating that the included covariates accounted for a modest proportion of the neighbourhood-level variance. However, the wide confidence interval suggests some uncertainty in this estimate. The MOR in the fully adjusted model remained similar at 1.32 (95% CI: 1.18 to 1.57), reinforcing the persistence of neighbourhood-level influences even after accounting for individual-level factors.

## Discussion

### Main findings

This study, utilising data from the 2016 South Africa Demographic and Health Survey (SADHS), reveals that nearly half (45.3%) of adults in South Africa experience multimorbidity, defined as the co-occurrence of two or more chronic

conditions. The prevalence of multimorbidity varied significantly across provinces, ranging from 35.6% in Limpopo to 52.1% in the Eastern Cape, highlighting substantial geographic disparities in chronic disease burden. These findings underscore the growing epidemic of multimorbidity in South Africa and its implications for public health.

Several key individual-level factors were significantly associated with an increased risk of multimorbidity. Older age emerged as the strongest predictor, with individuals aged 65–95 years having 11.6 times higher odds of multimorbidity compared to those aged 15–24 years. Women were nearly twice as likely as men to experience multimorbidity, reflecting well-documented gender disparities in chronic disease burden. Obesity was another critical factor, with obese individuals having 38% higher odds of multimorbidity compared to those with normal weight. Occupational exposure to smoke, dust, or fumes was also associated with a 26% increase in the odds of multimorbidity, emphasising the role of workplace hazards in chronic disease development. Additionally, individuals who were formerly married, had primary or secondary education, or consumed moderate amounts of sugary drinks were at higher risk of multimorbidity.

At the contextual level, neighbourhood and provincial factors played a significant role in shaping the distribution of multimorbidity. Neighbourhood-level poverty was associated with a reduced likelihood of multimorbidity, a finding that may reflect differences in healthcare access or health-seeking behaviours among residents of poorer neighbourhoods. Conversely, individuals residing in provinces with high levels of poverty were more likely to be multimorbid, suggesting that broader socioeconomic inequities contribute to chronic disease burden. Interestingly, provinces with higher unemployment rates and moderate rurality were associated with lower odds of multimorbidity, possibly due to differences in lifestyle factors or environmental exposures.

## Comparisons with previous studies

The high prevalence of multimorbidity found in this study (45.3%) is consistent with some previous estimates from South Africa, such as the 63.4% prevalence among older adults reported by Chang et al. (2019) [32] using data from the World Health Organization's Study on Global AGEing and Adult Health (SAGE). However, our estimate is higher than the 30.7% prevalence reported by Roomaney et al. (2022) [14] using the same SADHS 2016 dataset. This discrepancy may be attributed to differences in the number and type of chronic conditions included, as well as the handling of missing data. Our study's comprehensive inclusion of 12 chronic conditions and use of multiple imputation for missing data likely contributed to a more accurate estimation of the true multimorbidity burden in South Africa.

The strong association between older age and multimorbidity aligns with the global literature, which consistently identifies age as a major risk factor for chronic disease accumulation [33,34]. The gender disparity in multimorbidity, with women having nearly twice the odds of men, also echoes findings from other settings. For example, a systematic review by Violan et al. (2014) [35] found that female gender was associated with a higher prevalence of multimorbidity in most studies. This disparity may reflect a combination of biological, behavioural, and societal factors that shape women's health outcomes [36].

The significant association between obesity and multimorbidity is consistent with a growing body of evidence linking excess body weight to the development of multiple chronic conditions. The mechanisms underlying this relationship may involve chronic inflammation, metabolic dysregulation, and other pathophysiological processes [37,38].

The finding that occupational exposure to smoke, dust, or fumes increased the odds of multimorbidity by 26% is a novel contribution to the literature. While previous studies have documented the adverse health effects of workplace hazards on specific chronic conditions, such as respiratory diseases and cancer [39,40], few have examined their impact on multimorbidity. This underscores the importance of considering occupational factors in chronic disease research and prevention efforts.

The associations between socioeconomic factors and multimorbidity in our study were complex and sometimes counterintuitive. For example, we found that neighbourhood-level poverty was associated with a reduced likelihood of multimorbidity, while provincial-level poverty increased the odds. These findings contrast with some previous studies that have consistently linked lower socioeconomic status with higher multimorbidity prevalence [9,10]. However, other studies have reported mixed or null associations, suggesting that the relationship between socioeconomic factors and multimorbidity

may vary depending on the specific measures used and the context [4,41]. Our results highlight the need for further research to disentangle the complex interplay between individual and contextual socioeconomic determinants of multimorbidity in South Africa and other settings.

## Policy implications and implications for future research

The high prevalence of multimorbidity found in this study, affecting nearly half of the adult population in South Africa, underscores the urgent need for comprehensive policies and interventions to address this growing public health challenge. The findings highlight several key areas for policy action and future research.

First, the strong association between older age and multimorbidity emphasises the importance of developing targeted strategies to promote healthy aging and prevent chronic disease accumulation across the life course. This may include policies to support healthy lifestyles, such as physical activity promotion and nutrition education, as well as efforts to improve access to preventive healthcare services for older adults [42,43]. Future research should focus on identifying effective interventions to prevent and manage multimorbidity in older populations, taking into account their unique needs and preferences [44].

Second, the gender disparity in multimorbidity, with women being disproportionately affected, calls for gender-sensitive approaches to chronic disease prevention and management. This may involve policies to address the social, economic, and cultural factors that contribute to women's health disadvantages, such as gender-based violence, discrimination, and unequal access to education and employment opportunities [45]. Future research should explore the mechanisms underlying gender differences in multimorbidity and identify strategies to promote gender equity in health outcomes.

Third, the significant association between obesity and multimorbidity highlights the need for comprehensive policies to address the obesity epidemic in South Africa. This may include measures to promote healthy diets and physical activity, such as taxes on sugary beverages, regulations on food marketing, and investments in public infrastructure for active transportation [46,47]. Future research should evaluate the effectiveness of these policies in reducing obesity and its associated chronic disease burden, as well as explore innovative approaches to behaviour change and weight management.

Fourth, the finding that occupational exposure to smoke, dust, or fumes increased the odds of multimorbidity underscores the importance of strengthening workplace health and safety regulations in South Africa. This may involve policies to reduce exposure to hazardous substances, promote the use of personal protective equipment, and provide access to occupational health services [48]. Future research should investigate the specific occupational risk factors for multimorbidity in different industries and occupations, as well as evaluate the effectiveness of interventions to reduce workplace exposures and promote worker health [49].

Finally, the complex associations between socioeconomic factors and multimorbidity found in this study highlight the need for a nuanced and context-specific approach to addressing health inequities in South Africa. While some findings, such as the increased odds of multimorbidity in provinces with high poverty levels, align with the broader literature on social determinants of health [50], others, such as the reduced likelihood of multimorbidity in poorer neighbourhoods, challenge conventional wisdom. These results underscore the importance of considering multiple levels of influence on health outcomes, from individual to neighbourhood to provincial factors, and tailoring interventions accordingly [51]. Future research should aim to disentangle the causal pathways linking socioeconomic factors to multimorbidity in South Africa, as well as identify effective strategies for reducing health disparities across different contexts [52].

## Study strengths and limitations

This study has several notable strengths that contribute to its robustness and relevance for understanding the epidemiology of multimorbidity in South Africa. First, the use of nationally representative data from the 2016 South Africa Demographic and Health Survey (SADHS) ensures that the findings are generalisable to the adult population of the country. The large sample size and the inclusion of individuals from all nine provinces and across various sociodemographic groups enhance the external validity of the results.

Second, the comprehensive assessment of 12 chronic conditions, including both self-reported and objectively measured conditions, provides a more accurate estimation of the true burden of multimorbidity compared to studies that rely solely on self-report or a limited set of conditions [53]. The inclusion of biomarkers, such as blood pressure and glycated haemoglobin, reduces the potential for underdiagnosis and misclassification of conditions like hypertension and diabetes [54].

Third, the multilevel modelling approach used in this study allows for the simultaneous examination of individual and contextual factors associated with multimorbidity, providing a more nuanced understanding of the complex determinants of chronic disease burden [27–30]. By accounting for the clustering of individuals within neighbourhoods and provinces, the analysis reduces the potential for ecological fallacy and provides more accurate estimates of the associations between variables at different levels of influence [55].

However, the study also has some limitations that should be considered when interpreting the findings. First, the cross-sectional nature of the data precludes any causal inferences about the relationships between the significant variables and multimorbidity [56]. While the associations identified in this study are consistent with previous research and theoretical expectations, longitudinal studies are needed to establish the temporal sequence and causal pathways linking factors such as age, gender, obesity, and occupational exposures to the development of multiple chronic conditions over time [34].

Second, the reliance on self-reported data for some of the chronic conditions and risk factors may introduce recall bias or social desirability bias [57]. For example, individuals may underreport conditions that are stigmatised or not well-understood, such as mental health disorders, leading to an underestimation of their prevalence and association with multimorbidity [58]. Similarly, self-reported data on occupational exposures and health behaviours, such as smoking and alcohol consumption, may be subject to misclassification or underreporting.

Third, the study did not include some potentially important variables that have been associated with multimorbidity in other settings, such as family history, early life experiences, and healthcare access and utilisation [59,60]. The absence of these variables may lead to residual confounding and limit the ability to fully explain the observed associations between the significant variables and multimorbidity [61].

## Conclusion

This nationally representative study reveals a high burden of multimorbidity in South Africa, with nearly half of the adult population affected. Significant variations across provinces and sociodemographic groups were observed, with older age, female gender, obesity, occupational exposures, and certain neighbourhood and provincial characteristics emerging as key determinants. The findings underscore the urgent need for integrated, person-centred approaches to prevent and manage multiple chronic conditions in South Africa. Targeted interventions should address the needs of high-risk groups, while policies tackling the social determinants of health at the neighbourhood and provincial levels are crucial. Future research should employ longitudinal and qualitative methods to further elucidate the causal pathways and lived experiences of multimorbidity. By addressing the complex, multilevel factors driving the multimorbidity epidemic, South Africa can promote healthy aging, reduce health inequities, and improve population well-being.

## Supporting information

**S1 Table. Definitions of outcome measures.**
(DOCX)

**S2 Table. Information on explanatory variables and their derivation.**
(DOCX)

## Acknowledgments

The authors are grateful to Statistics South Africa (SSA), South African Medical Research Council (SAMRC), National Department of Health (NDoH) for providing the 2016 SADHS data for this analysis.

## Author contributions

**Conceptualization:** Andre Pascal Kengne, Olalekan Uthman.

**Data curation:** Matthew Hazell.

**Formal analysis:** Matthew Hazell.

**Methodology:** Matthew Hazell, Dylan Taylor, Olalekan Uthman.

**Supervision:** Olalekan Uthman.

**Writing – original draft:** Matthew Hazell.

**Writing – review & editing:** Matthew Hazell, Andre Pascal Kengne, Paramjit Gill, Dylan Taylor, Olalekan Uthman.

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
