## [Decision Letter · Decision Letter 0]

22 Oct 2024

PGPH-D-24-01547

A multilevel analysis of the prevalence and factors associated with multimorbidity in South Africa using 2016 Demographic and Health Survey data.

Dear Dr. Hazell,

Thank you for submitting your manuscript to PLOS Global Public Health. After careful consideration, we feel that it has merit but does not fully meet PLOS Global Public Health’s publication criteria as it currently stands. Therefore, we invite you to submit a revised version of the manuscript that addresses the points raised during the review process.

Please note that we have only been able to secure a single reviewer to assess your manuscript. We are issuing a decision on your manuscript at this point to prevent further delays in the evaluation of your manuscript. Please be aware that the editor who handles your revised manuscript might find it necessary to invite additional reviewers to assess this work once the revised manuscript is submitted. However, we will aim to proceed on the basis of this single review if possible. 

We look forward to receiving your revised manuscript.

Kind regards,

Vanessa Carels

Staff Editor

Journal Requirements:

1. Your current Financial Disclosure states, “MH (Pre-doctoral Fellowship; NIHR302654) is funded by the National Institute for Health Research (NIHR). The views expressed in this publication are those of the author(s) and not necessarily those of the NIHR, NHS or the UK Department of Health and Social Care.”. However, your funding information on the submission form is missing. Please indicate by return email the full and correct funding information for your study and confirm the order in which funding contributions should appear. Please be sure to indicate whether the funders played any role in the study design, data collection and analysis, decision to publish, or preparation of the manuscript.

3. Please provide separate figure files in .tif or .eps format.

4. We notice that your figures are included in the manuscript file. Please remove them as there are already an uploaded file. 

5. We have noticed that you have uploaded Supporting Information files, but you have not included a list of legends. Please add a full list of legends for your Supporting Information files after the references list. 

Additional Editor Comments (if provided):

Reviewers' comments:

Reviewer's Responses to Questions

**Comments to the Author**

1. Does this manuscript meet PLOS Global Public Health’s publication criteria ? Is the manuscript technically sound, and do the data support the conclusions? The manuscript must describe methodologically and ethically rigorous research with conclusions that are appropriately drawn based on the data presented.

Reviewer #1: Partly

2. Has the statistical analysis been performed appropriately and rigorously?

Reviewer #1: No

3. Have the authors made all data underlying the findings in their manuscript fully available (please refer to the Data Availability Statement at the start of the manuscript PDF file)?

Reviewer #1: No

4. Is the manuscript presented in an intelligible fashion and written in standard English?

Reviewer #1: Yes

5. Review Comments to the Author

Reviewer #1: First, I would like to thank the editorial board for the opportunity to review this article. I consider it to be a highly interesting work, wherein the authors estimate the prevalence and distribution of multimorbidity in South Africa, as well as determine the risk factors associated with the presence of multimorbidity. I find the use of multilevel models particularly intriguing, although I have some doubts, comments, and suggestions.

I am uncertain whether it is necessary to mention the type of adjusted model (multilevel analysis) in the title. Although multilevel analysis is not widely used, it is not a novelty either. This is, however, a purely personal opinion.

I couldn't analyze the bibliographic references because in the copy I received, they appear numbered in order of appearance within the article's text, while in the bibliography section they are listed alphabetically.

The introduction appears to be very well-written, clearly describing the objectives of the work and its potential implications.

MATERIALS AND METHODS

Study population

I suggest referencing Figure 1 (Study population).

Outcome variable

On page 7, lines 137-138, the 12 diseases from which multimorbidity is identified are mentioned (repetitively), having just been presented in the previous paragraph (page 6, lines 128-131).

On page 7, lines 139-141, the authors provide an example of a question and answer from the form. I find it more interesting to show the form in the supplementary material and reference it there, at least regarding the questions related to diseases.

On page 7, line 142, the authors state that if the interviewee responds that they don't know if they have a disease, this is interpreted as not having it. I find this criterion inappropriate. Rather, they should treat it as missing data, exactly as they do in the case of explanatory variables (page 8, lines 179-180). Moreover, the authors perform a sensitivity analysis with the entire dataset (with and without missing data), which, in my opinion, would make more sense if the main dataset consisted of individuals with complete data, and the sensitivity analysis with all individuals with imputation of missing data.

Statistical methods

In the first paragraph, the authors state that they use STATA software to conduct the statistical analysis. I recommend making this comment at the end of the statistical methodology section, not at the beginning.

At the end of page 9 and the beginning of page 10, the authors say they adjust 5 multilevel models. I recommend replacing the term "produced" with "fitted".

As the authors correctly mention, the adjustment of these 5 models, as generated, allows for explaining the effect of the three levels analyzed: individuals, neighborhoods, and provinces. However, it does not allow for an accurate analysis of which risk factors are most relevant, which is one of the study's objectives. For example, different models could be fitted, starting from a baseline model with age and sex, and then adding different groups of explanatory variables to contrast which are significant and which are not.

The final model contains 23 covariates with 50 associated coefficients. I consider that, without previously evaluating whether each and every variable is significant, there is a high risk of overfitting and multicollinearity. There are different techniques to reduce dimensionality, such as LASSO techniques, stepwise adjustment, or comparing Bayesian (BIC) or Akaike (AIC) criteria between nested models. I suggest applying some of these techniques to find which variables are most relevant and what their final effect is.

I have some doubts regarding the use of the multilevel model, particularly concerning the inclusion of the third level: province. Although I am not an expert in the matter, the multilevel methodology does not seem to apply at the provincial level. A typical example of applying multilevel techniques occurs when we have a sample of units at the level in question (in this case, provinces) and we are not interested in analyzing the differences between these units, but we understand that they may explain part of the total variability of the analysis (Harvey Goldstein. Multilevel Statistical Models. Fourth Edition. Page 17). However, in this work, the opposite situation occurs: information is available for all provinces of South Africa (9), and one of the objectives is to analyze the distribution of multimorbidity. Therefore, in my opinion, it would be more advisable to use the province as a fixed effect rather than a random effect. Moreover, the results, as the authors themselves comment, do not seem to support much the contribution of the provincial level. For all these reasons, I would recommend simplifying the multilevel model to two levels, individuals and neighborhoods, and using the province as a fixed effect.

RESULTS

Fixed effects (measures of association)

I suggest modifying the title of this section to "Measures of association: fixed effects". One of the objectives is to determine the risk factors for presenting multimorbidity. Therefore, I would give more relevance to this fact, rather than to the type of adjusted model: with fixed and random effects. Although I recognize this is a very personal opinion.

I find the writing of this section too dense, and in some cases, in unusual language. For example, on page 13 line 287, the authors say "Men had 0.54 times the odds of being multimorbid..." when it would be more comprehensible to say that the OR in men is 0.54, or that the probability of presenting multimorbidity is lower in men than in women (adding the values and/or confidence intervals in parentheses). Moreover, some of the results show very low significance, including unexpected or anomalous results, such as having a certain level of education being a risk factor compared to having no education. It is for this reason that I recommend adjusting different models, using dimension reduction techniques, to see which is the best final model, which variables have the most impact, and in what direction. My opinion is that with 23 covariates there may be high multicollinearity.

On the other hand, on page 13, line 290, they refer to the interpretation of age in terms of a 10-year increment, while Table 3 says that 5-year groups are used (not 10).

Table 1: characteristics of study population

I recommend adding a column with the p-values of the effect for each analyzed covariate.

I also recommend putting the percentages by rows and not by columns, this would allow seeing for each category the percentage of cases with multimorbidity. For example, in the case of education, it would be seen that the percentage of cases with multimorbidity in the group without studies is 65%, in the case of having primary studies the percentage with multimorbidity is 57%, and in the case of having secondary and higher studies the percentage is 39% and 34% respectively. This bivariate descriptive result would indeed show an expected result: the higher the level of education, the lower the presence of multimorbidity.

I also recommend putting all categories of each variable. For example, in the case of tobacco exposure at work, only the values of those who are exposed appear.

From the results, we can see that in some variables there is a high number of groups, some of which with few cases. It might be advisable to group categories, which would reduce the number of coefficients when adjusting the models.

Table 3: Multilevel logistic regression models

The results of Table 3 show that the effect of many of the variables introduced in the model is not significant, and in some cases the direction of the effect is surprising. Introducing variables that are not significant may be affecting the estimation of the rest. It is for this reason that I recommend reducing the dimension of the model.

A result that I find very interesting is that of tobacco exposure at work. If this effect is maintained by reducing the dimension of the model, it could be an interesting topic to address in the discussion, for example, to propose preventive measures.

I don't see in Table 3 the results of the variance at the individual level. Typically, if there are three levels, the results of the variance for each of them are shown. But Table 3 only shows the estimates of variances in two of the three levels (for neighborhood and province).

DISCUSSION

On page 18 lines 403-404 the authors say "The relationship between increasing education up to secondary level and reduced odds of multimorbidity..." when in my opinion it is the opposite (according to the multivariate model). In Table 3 the OR for secondary education is 1.31, and the reference category is cited as equivalent to having no studies. This is a surprising result, which is probably affected by multicollinearity and/or possible overfitting.

RECOMMENDATIONS

My main recommendation, oriented towards the objective of determining risk factors for presenting multimorbidity, would be to reduce the dimension of the final model. From the results of Table 3, it seems that some of the introduced variables are not significant. And given the high number of coefficients in the final model, the estimation of effects may be inefficient.

For this purpose, different techniques can be used, such as stepwise techniques, or the comparison of model performance based on measures that penalize the introduction of additional coefficients: such as the BIC.

I also recommend introducing the effect of provinces as a fixed effect, and reducing the random effects to individuals and neighborhoods.

6. PLOS authors have the option to publish the peer review history of their article (what does this mean? ). If published, this will include your full peer review and any attached files.

**Do you want your identity to be public for this peer review?** For information about this choice, including consent withdrawal, please see our Privacy Policy .

Reviewer #1: **Yes: ** David Monterde

---

## [Decision Letter · Decision Letter 1]

18 Mar 2025

PGPH-D-24-01547R1

Prevalence, Patterns, and Determinants of Multimorbidity in a Resource-Limited Setting: Insights from a Nationally Representative Survey

Dear Dr. Hazell,

Thank you for submitting your manuscript to PLOS Global Public Health. After careful consideration, we feel that it has merit but does not fully meet PLOS Global Public Health’s publication criteria as it currently stands. Therefore, we invite you to submit a revised version of the manuscript that addresses the points raised during the review process.

We look forward to receiving your revised manuscript.

Kind regards,

Prof Razak Gyasi, PhD, PD

Academic Editor

Journal Requirements:

Additional Editor Comments (if provided):

As you will see, the reviewers expressed some important minor concerns. Therefore, I am unable to accept your paper in its current form but invite you to respond to the reviewer comments and revise your manuscript accordingly. Please consider the grammar and language structure edits throughout the draft.

Reviewers' comments:

Reviewer's Responses to Questions

**Comments to the Author**

1. If the authors have adequately addressed your comments raised in a previous round of review and you feel that this manuscript is now acceptable for publication, you may indicate that here to bypass the “Comments to the Author” section, enter your conflict of interest statement in the “Confidential to Editor” section, and submit your "Accept" recommendation.

Reviewer #1: All comments have been addressed

2. Does this manuscript meet PLOS Global Public Health’s publication criteria ? Is the manuscript technically sound, and do the data support the conclusions? The manuscript must describe methodologically and ethically rigorous research with conclusions that are appropriately drawn based on the data presented.

Reviewer #1: Yes

3. Has the statistical analysis been performed appropriately and rigorously?

Reviewer #1: Yes

4. Have the authors made all data underlying the findings in their manuscript fully available (please refer to the Data Availability Statement at the start of the manuscript PDF file)?

Reviewer #1: Yes

5. Is the manuscript presented in an intelligible fashion and written in standard English?

Reviewer #1: Yes

6. Review Comments to the Author

Reviewer #1: The authors have responded to all my comments and suggestions.

All my suggestions have been taken into account and either carried out, or the reason for not doing so has been perfectly and exhaustively justified.

Reading the updated version, I only detected one minor error: on page 11, line 251, the authors say "Women had significantly higher odds of multimorbidity compared to women", when I understand it should say "Women had significantly higher odds of multimorbidity compared to men".

I would like to thank the authors for their efforts in carrying out all the modifications.

I have no further comments.

Congratulations on the work done.

7. PLOS authors have the option to publish the peer review history of their article (what does this mean? ). If published, this will include your full peer review and any attached files.

**Do you want your identity to be public for this peer review?** For information about this choice, including consent withdrawal, please see our Privacy Policy .

Reviewer #1: **Yes: ** David Monterde

---

## [Editor Report · Decision Letter 2]

1 Apr 2025

PGPH-D-24-01547R2

Prevalence, Patterns, and Determinants of Multimorbidity in a Resource-Limited Setting: Insights from a Nationally Representative Survey

Dear Dr. Hazell,

Thank you for submitting your manuscript to PLOS Global Public Health. After careful consideration, we feel that it has merit but does not fully meet PLOS Global Public Health’s publication criteria as it currently stands. Therefore, we invite you to submit a revised version of the manuscript that addresses the points raised during the review process.

We look forward to receiving your revised manuscript.

Kind regards,

Professor Razak Gyasi, PhD, PD

Academic Editor

Journal Requirements:

Additional Editor Comments (if provided):

Many thanks for implementing all comments from our experienced reviewers. Please address the following issues so that we can progess the manuscript for accetance.

**1. The Abstract must be structured and conceptually divided into sections, including Background, Methodology, Principal Findings, and Conclusions/Significance. Please consider this in the manuscript.**

**2. The title of the paper should contain a specific context where the study was conduted. A resource-limited setting appears too broad and should replaced by a specific context.**
---

## [Editor Report · Decision Letter 3]

16 Apr 2025

Prevalence, Patterns, and Determinants of Multimorbidity in South Africa: Insights from a Nationally Representative Survey

PGPH-D-24-01547R3

Dear Mr Hazell,

We are pleased to inform you that your manuscript 'Prevalence, Patterns, and Determinants of Multimorbidity in South Africa: Insights from a Nationally Representative Survey' has been provisionally accepted for publication in PLOS Global Public Health.

Best regards,

Professor Razak Gyasi, PhD, PD

Academic Editor